# Cutting care clusters: the creation of an inverse pharmacy care law? An area-level analysis exploring the clustering of community pharmacies in England

Adam Todd,[1,2,3] Katie Thomson,[2,3] Adetayo Kasim,[4] Clare Bambra[2]

This article has received a badge for Open data.

¹School of Pharmacy, Newcastle University, Newcastle upon Tyne, UK
²Institute of Health and Society, Newcastle University, Newcastle upon Tyne, UK
³Fuse – the UKCRC Centre for Translational Research in Public Health, Newcastle upon Tyne, UK
⁴Wolfson Research Institute for Health and Wellbeing, Durham University, Stockton-on-Tees, UK

**Correspondence to**
Dr Adam Todd;
adam.todd@newcastle.ac.uk

## ABSTRACT

**Objectives** To (1) explore the clustering of community pharmacies in England and (2) determine the relationship between community pharmacy clustering, urbanity and deprivation.

**Design** An area-level analysis spatial study.

**Setting** England.

**Primary and secondary outcome measures** Community pharmacy clustering determined as a community pharmacy located within 10 min walking distance to another community pharmacy.

**Participants** Addresses and postal codes of each community pharmacy in England were used in the analysis. Each pharmacy postal code was assigned to a lower layer super output area, which was then matched to urbanity (urban, town and fringe or village, hamlet and isolated dwellings) and deprivation decile (using the Index of Multiple Deprivation score).

**Results** 75% of community pharmacies in England were located in a 'cluster' (within 10 min walking distance of another pharmacy): 19% of community pharmacies were in a cluster of two, while 56% of community pharmacies were in clusters of three or more. There was a linear relationship between community pharmacy clustering and social deprivation—with clustering more prevalent in areas of higher deprivation: for community pharmacies located in areas of lowest deprivation (decile 1), there was a significantly lower risk of clustering compared with community pharmacies located in areas of highest deprivation (relative risk 0.12 (95% CI 0.10 to 0.16)).

**Conclusions** Clustering of community pharmacies in England is common, although there is a positive trend between community pharmacy clustering and social deprivation, whereby clustering is more significant in areas of high deprivation. Arrangements for future community pharmacy funding should not solely focus on distance from one pharmacy to another as means of determining funding allocation, as this could penalise community pharmacies in our most deprived communities, and potentially have a negative effect on other healthcare providers, such as general practitioner and accident and emergency services.

## Strengths and limitations of this study

- ► This is the first study exploring community pharmacy clustering, and how this varies according to social deprivation and urbanity.
- ► We conceptualised a community pharmacy cluster using a 0.5-mile straight line which is a limitation of the work.
- ► We did not model what would happen to community pharmacy access if the clusters were removed due to community pharmacy closures.

## INTRODUCTION

In recent years, community pharmacies have emerged as strategically important settings, which play a crucial role in delivering healthcare and public health services.[1] In the UK, for example, community pharmacies offer a range of tiered services: the first level of service, the essential service, includes dispensing medication and providing medication-related advice; the second, the advanced service, allows pharmacists, among other things, to provide influenza vaccinations and medicines use reviews; the third, the locally commissioned services are used to address the needs of the local population and can include smoking cessation, and minor illness schemes.[2] Importantly, through these patient-focused roles, community pharmacy services are able to manage people with minor illnesses and some long-term conditions. This is advantageous, as people with minor illness can be directed away from other healthcare providers, such as general practitioners (GPs) and accident and emergency (A&E) departments.[3 4] This approach, therefore, has the potential to free up valuable healthcare resource, which allows other primary and secondary care services to mange people with more serious conditions where community pharmacy involvement would not be appropriate.

Indeed, with the average waiting times for a non-urgent GP appointment around 13 days, and the number of people waiting longer than 4 hours in A&E departments

increasing, it is clear UK healthcare services are under unprecedented strain.[5] Furthermore, it is known that people living in areas of higher socioeconomic deprivation are more likely to need primary and secondary health services, but that provision is in fact lower in these high-need neighbourhoods: the inverse care law.[6] The problem of timely access to healthcare services is therefore adversely affecting people in the most deprived communities—potentially further widening health inequalities and thereby putting even more pressure on healthcare services in the longer term. Previous work has shown that, in contrast to the inverse care law for GPs and hospital services, a positive pharmacy care law exists—whereby people living in areas of highest deprivation have better access to community pharmacies.[7] It is also known that the urbanity of an area is an important consideration for healthcare access, with people from urban areas living in greater proximity to GP services, when compared with people living in rural areas.[8] Community pharmacies, therefore, have the potential to reach people in the areas of greatest need thereby offering additional healthcare access in areas that are traditionally 'underdoctored'— including rural areas.[8]

Despite the significant—and perhaps underused— potential of community pharmacies, austerity has seen the implementation of UK government funding cuts to the English community pharmacy sector—with some estimates suggesting one in four community pharmacies could close as a result of the reduced funding envelope.[9] It has been argued that because community pharmacies cluster together, some can be cut without impacting on service provision. In a letter to the Pharmaceutical Services Negotiating Committee (the body that represents National Health Service (NHS) pharmacy contractors in England), the Department of Health state that '40% of pharmacies are in a cluster where there are three or more pharmacies within 10 min walk' and that 'in some parts of the country there are more pharmacies than are necessary to maintain good access'.[10] Despite these claims, and the potential impact on funding allocations and therefore service provision, there are no published studies that explore the clustering of community pharmacies in England, and how such clustering is linked to socioeconomic deprivation and urbanity. The research, therefore, aimed to: (1) explore the clustering of community pharmacies in England by 10 min walking distance and (2) determine the relationship between community pharmacy clustering, urbanity and deprivation.

## METHODS

### Design

This study used geographical information systems to explore community pharmacy clustering according to urbanity and deprivation. For the purposes of the study, a community pharmacy was defined as a premises registered with the General Pharmaceutical Council for the purposes of compounding, procurement, storage and distribution of medicines and appliances[11]; we excluded premises that were solely registered as internet pharmacies in the analysis.

### Data and variables

Community pharmacy data were obtained from the Geo-Healthcare Access Database.[12] This open access database contains data on the address and postal code of each community pharmacy premises in England (matched to their corresponding coordinates using the Office of National Statistics postcode directory, 2014); the last update of the dataset was in 2016. Community pharmacy locations were mapped and their corresponding 2011 lower layer super output areas (LSOAs) were extracted. There are 32 844 LSOAs in England and these geographical areas comprise approximately 1–3000 people living in 400–1200 households, and were designed to improve the reporting of small area statistics. Urbanity is also a factor in access to healthcare services including community pharmacies so, using the rural–urban classification (2011) each community pharmacy LSOA was assigned to one of three categories: (1) urban, (2) town or fringe or (3) village, hamlet and isolated dwellings. These categories were aggregated from the original rural–urban classification which assigns areas to one of six rural or four urban settlement/ context types. The 2015 Index of Multiple Deprivation (IMD) score was also matched to each community pharmacy's LSOA (from the Office of National Statistics). The IMD is an overall measure of multiple deprivation experienced by people living in an area, and comprises 37 separate indicators organised across seven domains of deprivation (income, employment, health and disability, education, skills and training, crime, barriers to housing and services and living environment) which are combined, using appropriate weights.[13]

### Data analysis

Locations of community pharmacies in England were mapped using ArcMap (V.10.3) and 0.5-mile (straight-line) buffers were placed around the site to represent a 10 min walk, using an average walking speed of 3 mph (4 km).[14] The number of community pharmacies which were isolated or clustered in groups of two, or three or more were then extracted for all locations. A cluster was defined if the buffer around a single community pharmacy contained two or more unique sites. The number of community pharmacies within each individual buffer was then calculated and summarised. In addition to examining data for all community pharmacies, further breakdown of clustering took place depending on whether or not the LSOA of the pharmacy was 'urban', 'town or fringe' or 'village, hamlet and isolated dwellings'. This process was repeated for all deprivation deciles based on the IMD 2015 for England, whereby the most deprived decile (decile 10) equated to the most deprived 10% of LSOAs, while the least deprived

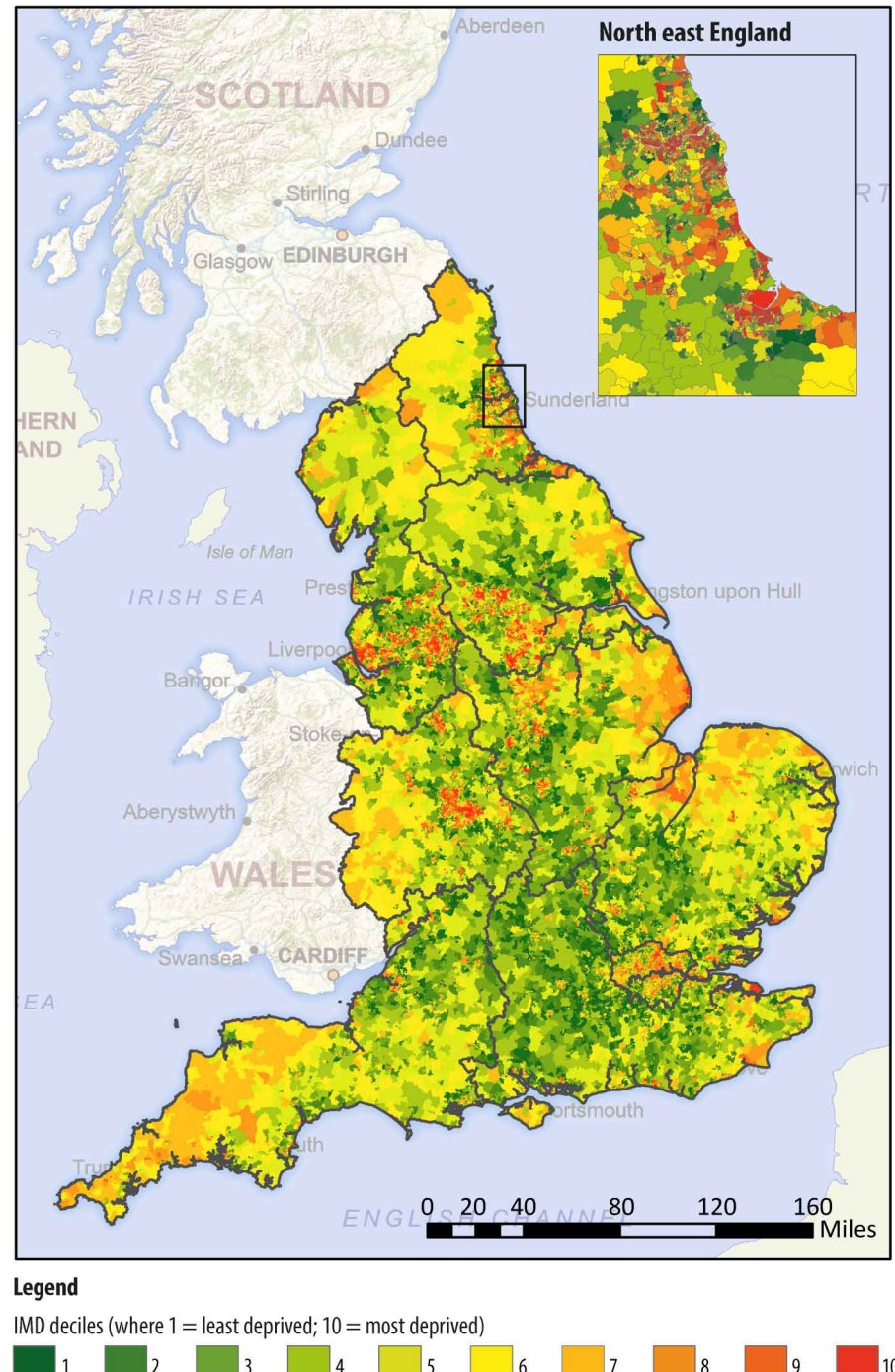

**Legend**

IMD deciles (where 1 = least deprived; 10 = most deprived)

| 1 | 2 | 3 | 4 | 5 | 6 | 7 | 8 | 9 | 10 |

**Figure 1** Map of England with LSOAs stratified according to deprivation. IMD, Index of Multiple Deprivation; LSOAs, lower layer super output areas.

decile (decile 1) represented the 10% least deprived LSOAs (figure 1). The relative risk for community pharmacy clustering by deprivation was then calculated according to deprivation decile. For the deprivation analysis, a community pharmacy 'cluster' was considered as three or more community pharmacies within a 10 min walking distance of each other.

### Patient and public involvement

As this study involved secondary data analysis from the Geo-Healthcare Access Database, patients or the public were not involved in the design or delivery of this research.

### RESULTS

Overall, our results show that the percentage of community pharmacies within 10 min walking distance (0.5 mile) of one another is 75%: 19% of community pharmacies were in a cluster of two, while 56% of community pharmacies were in clusters of three or more. An example of community pharmacy clustering is shown visually in figure 2.

### Clustering of community pharmacies by urban–rural classification

For community pharmacies located in urban areas (n=10 438), there was no clustering for 19% of community

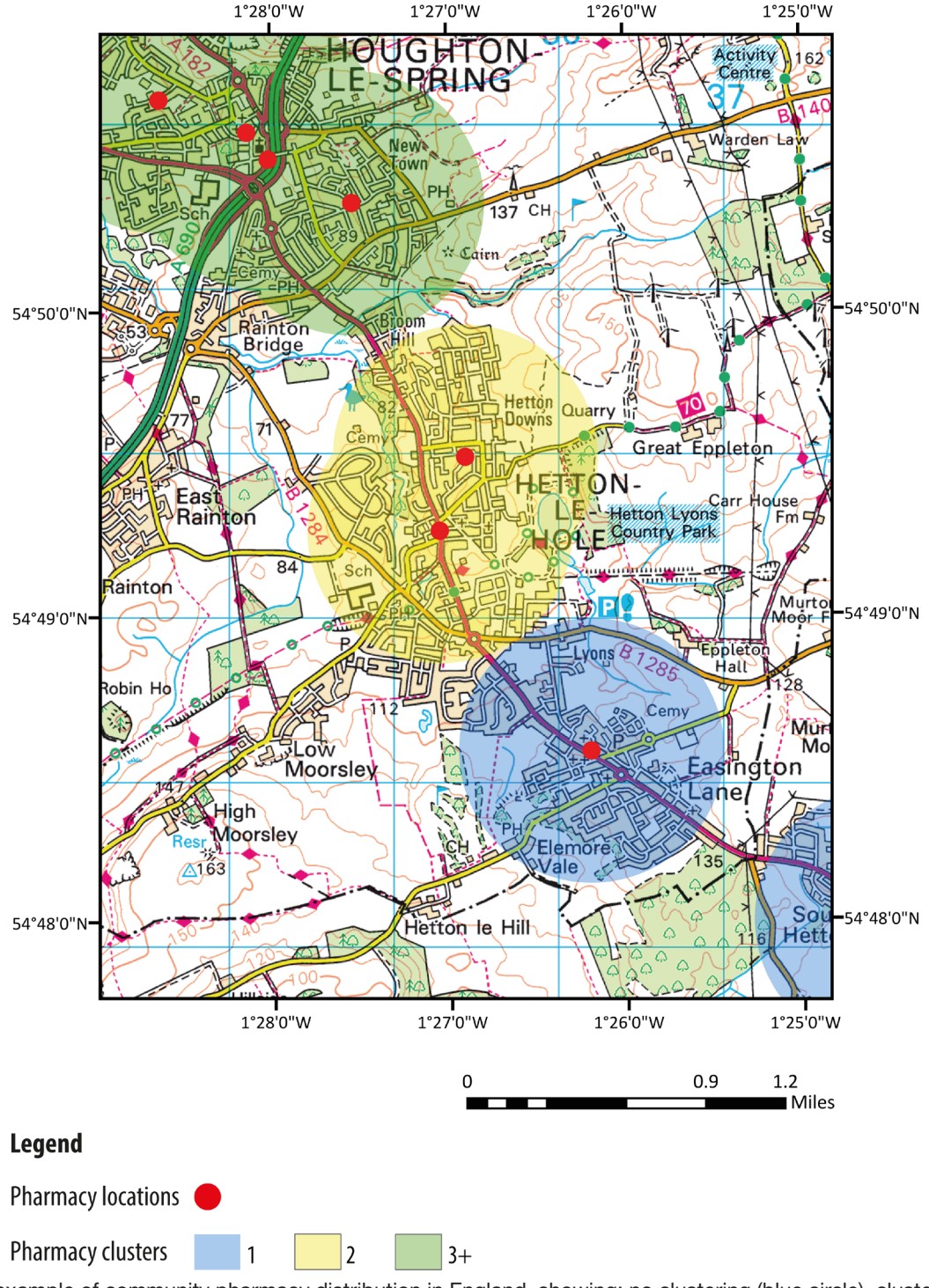

**Legend**

Pharmacy locations ⬤

Pharmacy clusters ▨ 1   ▨ 2   ▨ 3+

**Figure 2** An example of community pharmacy distribution in England, showing: no clustering (blue circle), clusters of 2 (yellow circle) and clusters of 3 or more (green circle).

pharmacies, 19% were located in a cluster of two, while 62% were located in clusters of three or more. In town and fringe areas (n=1147), there was no clustering in 94% of community pharmacies, 5% were located in clusters of two, while 1% were located in clusters of three or more. In village areas (n=153), there was no clustering in 94% of community pharmacies, 4% were located in clusters of two, while 2% were located in clusters of three or more.

**Clustering of community pharmacies by IMD**

When stratifying by IMD, there was, overall, a linear relationship between community pharmacy clustering and social deprivation—with clustering more prevalent in areas of higher deprivation (figure 3); the highest percentage of community pharmacy clustering was 62%, observed in deprivation decile 10, while the lowest percentage of community pharmacy clustering was 8%,

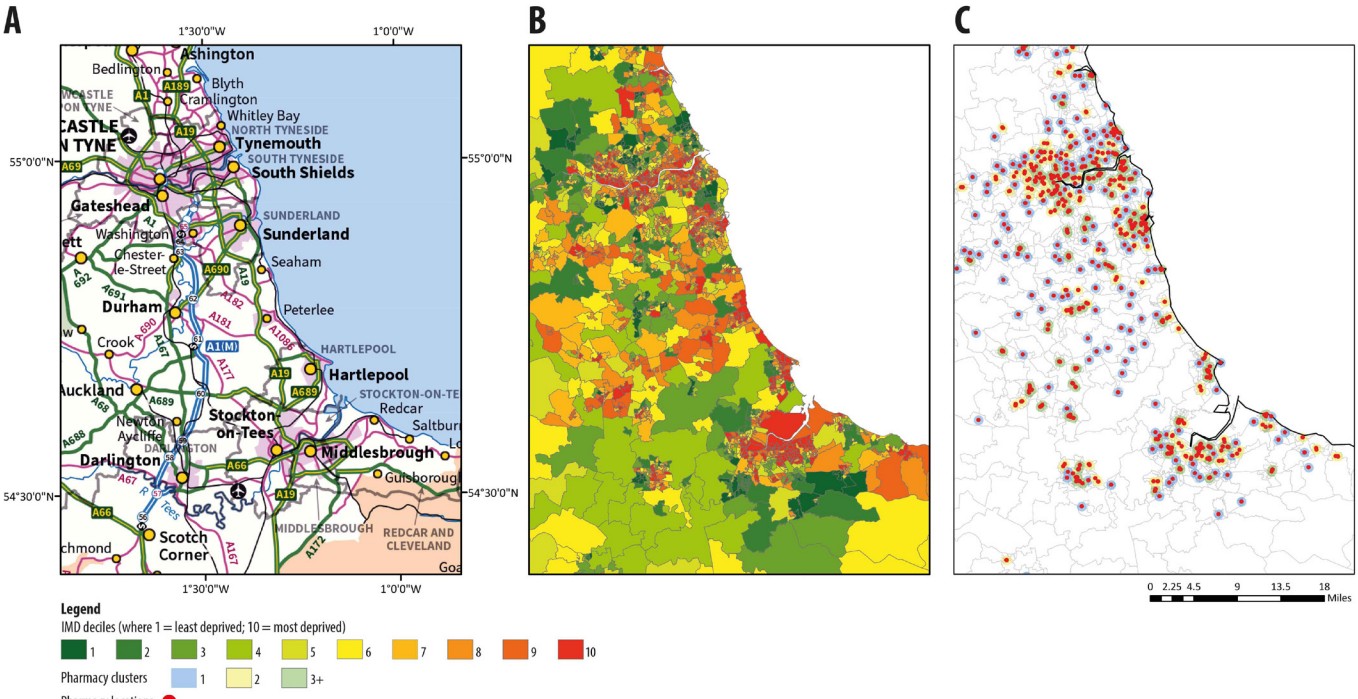

**Figure 3** Map of the Northeast of England showing; (A) ordinance survey map; (B) deprivation by LSOA and, (C) community pharmacy clustering. IMD, Index of Multiple Deprivation; LSOAs, lower layer super output areas.

observed in deprivation decile 1 (figure 4). Community pharmacies located in the most deprived areas were significantly more likely to exist as clusters: for community pharmacy clusters of three or more, there was a significantly higher risk of clustering in deprivation decile 10, compared with all the other deciles. When comparing community pharmacies in decile one (least deprived) to community pharmacies in decile 10 (most deprived), there was an 88% lower risk of clustering for community pharmacies in the least deprived areas (table 1).

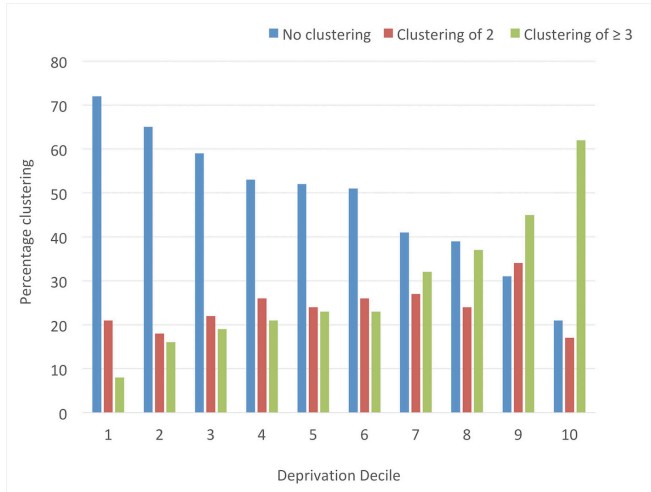

**Figure 4** Percentage clustering of community pharmacies in England according to social deprivation (1 is the least deprived, while 10 is the most deprived).

## DISCUSSION

The key findings of the study show that clustering of community pharmacies in England is common—with around 56% of all community pharmacies existing in a cluster of three or more. There is also a positive trend between deprivation and community pharmacy clustering, whereby clustering is significantly more common in areas of high deprivation. Clustering of community pharmacies was also more common in urban areas, when compared to rural areas. These findings may well reflect both health need and population size in the respective neighbourhoods.

This is the first study that has empirically explored whether community pharmacies are clustered in England—and how such clustering varies according to deprivation and urbanity. Previous literature has shown that geographical proximity of services can be an important consideration for healthcare utilisation; for example, Turnbull *et al* showed that call rates to primary care centres decreased with increasing distance. The same study also showed that higher call rates were associated with more deprived areas.[15] Furthermore, Lin *et al* who examined travel distance to hospital and the associated effect on hospitalisations in Canada, showed that admission rates were inversely proportional to hospital distance.[16] It has also been shown in the literature that proximity to healthcare services is an important consideration for healthcare utilisation, and is an important factor for health outcome. For example, a study by Okwaraji *et al* showed that, in a remote area of rural Ethiopia, the distance to the nearest health centre had a significant effect on child mortality—with children living greater

**Table 1** Relative risk (RR) of community pharmacy clustering according to deprivation decile (1 is the least deprived, while 10 is the most deprived)

| Deprivation decile | No clustering RR (95% CI) | Clustering of 2 RR (95% CI) | Clustering of ≥3 RR (95% CI) |
|---|---|---|---|
| 1 | 3.38 (3.06 to 3.72) | 1.20 (1.01 to 1.43) | 0.12 (0.10 to 0.16) |
| 2 | 3.08 (2.78 to 3.39) | 1.06 (0.89 to 1.26) | 0.27 (0.23 to 0.31) |
| 3 | 2.78 (3.51 to 3.08) | 1.28 (1.09 to 1.49) | 0.31 (0.27 to 0.35) |
| 4 | 2.52 (2.27 to 2.80) | 1.49 (1.29 to 1.72) | 0.34 (0.30 to 0.38) |
| 5 | 2.48 (2.23 to 2.74) | 1.40 (1.21 to 1.62) | 0.38 (0.34 to 0.42) |
| 6 | 2.41 (2.17 to 2.67) | 1.50 (1.32 to 1.73) | 0.37 (0.33 to 0.42) |
| 7 | 1.93 (1.73 to 2.15) | 1.53 (1.34 to 1.75) | 0.53 (0.49 to 0.56) |
| 8 | 1.83 (1.65 to 2.04) | 1.39 (1.22 to 1.58) | 0.60 (0.56 to 0.65) |
| 9 | 1.47 (1.31 to 1.64) | 1.40 (1.23 to 1.59) | 0.73 (0.68 to 0.78) |
| 10 (reference) | 1 | 1 | 1 |

than 1.5 hours walk from the health centre having a two to three greater risk of death compared with children who live within 1.5 hours walk.[17] Clearly the context of this work is different to healthcare settings in England, so limited comparisons can be drawn. However, given the well-established relationship between health and place, it would be prudent for future research to explore how community pharmacy distribution is associated with health of a particular area. The literature has also shown that healthcare access, from a geographical perspective, tends to be lower among rural communities when compared to urban communities. For example, Lovett *et al*, who explored GP accessibility by car travel and public transport, showed examples of rural areas in East Anglia (typically with low levels of personal mobility and high health need) where there was no daytime bus services or no community transport, which allowed travel/access to a GP.[18] These challenges have been summarised by Baird and Wright, who coined the term 'rural health deprivation', and argue that, to improve the health of the nation, more needs to be done to develop care pathways for rural communities.[19] Although we have not measured accessibility of community pharmacy by population, our findings that community pharmacy clustering is lower in rural areas lend support to the hypothesis that accessibility to healthcare is lower among rural communities.

In terms of study limitations, while we believe our results are robust, and have important policy implications for the way in which community pharmacies are funded, we acknowledge there are several: first, we recognise that a 10 min walk (0.5 mile) from each community pharmacy was represented using a straight-line distance from the central point of each community pharmacy's postal code to create a buffer. This assumes people are able to walk in any direction from that postal code and always in a straight line, while, in reality, people are often constrained to pathways that curve, or are cut-off by barriers (such as lakes, train tracks or busy roads). We also acknowledge that we did not consider

community pharmacy utilisation in our analysis; just because a community pharmacy is located in a particular area, does not necessarily mean people from that area choose to use it. In addition, we did not consider which types of service were being used from the community pharmacies; it is possible that service utilisation will change according to health need of the population (eg, people living in areas with a high prevalence of smoking might use more community pharmacy smoking cessation services). We also acknowledge that we did not explore how community pharmacy clustering varied according to population type; for example, it is possible that some community pharmacies serve different types of populations, such as black and minority ethnic groups or older populations. It would be prudent, therefore, that future work establishes the types of people that use community pharmacies services, and how this varies according to urbanity, deprivation and local health need.

The policy implications of this work are clear: cutting the funding of community pharmacies by clustering will disproportionately affect community pharmacies located in the most deprived communities: potentially leading to an inverse pharmacy care law. Given that community pharmacies provide effective public health services,[20] this has the potential to further widen health inequalities—should community pharmacies in these areas close. Indeed, it would undermine the department of health's responsibilities to reduce inequalities in access to NHS services. There have been recent reports of some large community pharmacy organisations taking the decision to close some of their community pharmacies as they have become 'commercially unviable', although it is not yet clear what areas will be affected.[21] We note that the department of health has introduced a Pharmacy Access Scheme (PhAS), whereby, according to specific criteria, certain community pharmacies will be protected from the reduction in funding.[22] Initially, community pharmacies would not qualify for the scheme if they were located within a 1-mile radius of another community pharmacy,

although we note that, in view of our preliminary work around community pharmacy clustering, the formula was modified to accommodate community pharmacies located in top 20% of deprived areas in England within a 0.8-mile radius of another community pharmacy.[23] At present, however, it is not clear if community pharmacy clustering is driven by health need of the local area, or, as many community pharmacies are located in urban areas, they benefit from the increased footfall by being located in these areas. Future work should seek to address this important issue. While we are encouraged by the concept of the PhAS recognising community pharmacies in deprived areas, we believe that future models of funding should be based on quality metrics around the provision of healthcare and public health services, rather focus on distance or clustering of pharmacies. The department of health has also recently introduced a quality payment scheme whereby extra payments will be paid to community pharmacies if certain quality criteria are met. We believe this is significant progress, and it is important that this concept continue which will remove focus on the number of services a community pharmacy undertakes, to that of the quality of the service a community pharmacy undertakes. This approach is critical, given that a recent study highlighted that community services are often not related to need of the local population, and can be influenced by pharmacy ownership type.[24] The development of a quality metric will help direct funding to community pharmacies that are engaged in delivering the higher levels of service.

In terms of the impact on the wider healthcare community, previous work shows that if community pharmacy services were not present, people would use alternate—and more costly—branches of the NHS, such as GP and A&E services.[3] It is also evident that people living in more deprived areas experience higher morbidity much earlier in life, compared to people living in affluent areas—as represented by lower healthy life expectancy.[25] A study analysing one million GP consultations in London showed that someone aged 50 years living in the most deprived quintile of English neighbourhoods consults with their GP at the same rate as someone aged 70 years in the most affluent quintile of neighbourhoods.[26] So, community pharmacy closures—particularly in deprived areas—are likely to increase the workload of general practice. Community pharmacies are able to reach people that other services cannot. For example, a study has shown that owing to their wider accessibility (eg, proximity, opening times, no appointment required), many people prefer to obtain healthcare from a community pharmacy setting[27] and a qualitative study revealed some people were more likely to use a community pharmacy for healthcare advice owing to their trusting relationship they develop with the pharmacist.[28]

## CONCLUSION

Just over half of community pharmacies in England are located in clusters of three or more by 10 min walking distance. There is also a positive trend between community pharmacy clustering and social deprivation, whereby clustering is significantly more common in areas of high deprivation. As such, arrangements for future community pharmacy funding should not solely focus on distance from one pharmacy to another, as means of determining funding allocation, as this could penalise community pharmacies in our most deprived communities, and potentially have a negative effect on other healthcare providers, such as GP and A&E services by increasing their workload. Future funding models of community pharmacy should consider quality of healthcare and public health services provided, as well as health need of the local area.

**Correction notice** The funding statement has been updated since publication and this article has changed to CC BY license

**Acknowledgements** CB is a member of Fuse (funding reference MR/K02325X/1). Funding for Fuse comes from the British Heart Foundation, Cancer Research UK, Economic and Social Research Council, Medical Research Council, the National Institute for Health Research, under the auspices of the UK Clinical Research Collaboration, and is gratefully acknowledged (RF150334).

**Contributors** CB and AT designed the study, led on data interpretation and supervised all stages of the research. KT led on the GIS modelling and AK conducted statistical analysis. AT led the drafting of the manuscript with input from CB, KT and AK. AT is guarantor of the work.

**Funding** CB is a member of Fuse (funding reference MR/K02325X/1). Funding for Fuse comes from the British Heart Foundation, Cancer Research UK, Economic and Social Research Council, Medical Research Council, the National Institute for Health Research, under the auspices of the UK Clinical Research Collaboration, and is gratefully acknowledged (RF150334)

**Disclaimer** The views expressed in this paper do not necessarily represent those of the funders or UKCRC.

**Competing interests** None declared.

**Patient consent** Not required.

**Ethics approval** Ethical approval was not required for this work as this study used non-patient identifiable secondary data; patients were not actively involved in this research.

**Provenance and peer review** Not commissioned; externally peer reviewed.

**Data sharing statement** All of our raw data are available through the open access Geo-healthcare database: http://collections.durham.ac.uk/files/fq977t77k#.WJSlsRicZBw

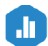

**Open data** All data have been made publicly available and can be accessed at http://collections.durham.ac.uk/files/fq977t77k#.WJSlsRicZBw

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
