## [Reviewer comments · BMJ Open]

ARTICLE DETAILS

TITLE (PROVISIONAL)	Cutting care clusters: the creation of an inverse pharmacy care law? An area level analysis exploring the clustering of community pharmacies in England
AUTHORS	Todd, Adam; Thomson, Katie; Kasim, Adetayo; Bambra, Clare

VERSION 1 – REVIEW

REVIEWER	Sally Jacobs The University of Manchester, UK
REVIEW RETURNED	12-Mar-2018

GENERAL COMMENTS	Many thanks for allowing me the opportunity to review this paper. It is an important and timely piece of work given current UK policy and cuts to pharmacy funding and provides evidence for the potential impact, particularly in deprived areas, of pharmacy closures on the basis of proximity to other pharmacies. The paper is well written and based on a simple but seemingly rigorous analysis of existing data. I do not have the necessary expertise to comment in detail on the statistical analysis but it appears to be an appropriate way of meeting the research objectives and the conclusions are clearly linked to the study findings. The findings contribute to the evidence that a positive care law exists for community pharmacy services. This is of particular interest to my own research which has recently demonstrated that the volume of services provided by individual pharmacies is inversely related to local population health need. The authors may wish to consider Hann et al. BMJ Open 2017;7:e017843 in their discussion. It would be interesting to extend and combine both pieces of analysis to explore whether or not local health needs are being met by community pharmacies given both sets of findings. I recommend that you accept this paper for publication following specialist statistical review.
---

REVIEWER	Jan Bauer Goethe University Frankfurt, Germany
REVIEW RETURNED	30-Mar-2018

GENERAL COMMENTS	This manuscript examined clustering of pharmacies in England and showed that clustering exists and varies geographically and there is a positive association between clustering and social deprivation. The manuscript is well written and easy to read. However, there are some concerns regarding the methods and the context of this manuscript. Please see the comments below.
---

	Introduction: The authors describe the potential of community pharmacies to fill the gap of GPs and AE departments in rural areas. However, there are obvious limitations for pharmacies filling that gap, which have not been addressed by the authors. The authors want to explore the relationship of clustering with urbanity. However is not explained why this has to be done. Please provide arguments as to why urbanity has to be included in the analysis. The authors mention the “positive care law” of pharmacies (access is highest in areas with highest deprivation). Later the authors declare that no study exist that examines the relationship of clustering and social deprivation. Access to healthcare can be measured in many different ways, also possibly by analyzing clusters. Therefore it is unclear if “clustering” is a measure of access. If so, please indicate this and provide context of other accessibility measures. If not please describe the difference between “clustering” and access and why this has to be analysed giving the already mentioned positive care law. Methods: The authors want to “explore geographical access to community pharmacies”. However, again it is not clear what to authors mean by “access” (see also comment in the introduction) and how they want to measure access. Please provide the year of the community pharmacy data. The authors used a 0,5 mile straight line buffer, which was supposed to represent “walking distance”. Using straight line buffers to measure walking distance is not state of the art in accessibility measures. State of the art would be to use walking speed on a street network. It is unclear why the authors have not used a street network. Therefore the straight line buffer used should not be referred to as “10 minute walking distance” but to “0,5 mile radius” which is more precise and less misleading. It is also unclear why the authors defined a pharmacy cluster as 2 or more pharmacies within the buffer. Please explain the choice of “2 or more”. It is also unclear why, for the deprivation analyses “3 or more” have been used for the definition of clusters. Discussion: Within the discussion the authors focused mainly on the possible implications. The actual discussion of the result in regard to other studies is almost completely missing. Furthermore putting the results in an international perspective would further improve this manuscript. References: Almost 50% of the presented literature (regarding journal articles) are form the authors themselves. Given the topic of this manuscript and the vast (international) available literature, the choice of the literature seems biased.
--	---

VERSION 1 – AUTHOR RESPONSE

Reviewer(s)' Comments to Author:

Reviewer: 1

Reviewer Name: Sally Jacobs

Institution and Country: The University of Manchester, UK

Please state any competing interests or state 'None declared': None declared

Please leave your comments for the authors below

Many thanks for allowing me the opportunity to review this paper. It is an important and timely piece of work given current UK policy and cuts to pharmacy funding and provides evidence for the potential impact, particularly in deprived areas, of pharmacy closures on the basis of proximity to other pharmacies. The paper is well written and based on a simple but seemingly rigorous analysis of existing data. I do not have the necessary expertise to comment in detail on the statistical analysis but it appears to be an appropriate way of meeting the research objectives and the conclusions are clearly linked to the study findings.

The findings contribute to the evidence that a positive care law exists for community pharmacy services. This is of particular interest to my own research which has recently demonstrated that the volume of services provided by individual pharmacies is inversely related to local population health need. The authors may wish to consider Hann et al. BMJ Open 2017;7:e017843 in their discussion. It would be interesting to extend and combine both pieces of analysis to explore whether or not local health needs are being met by community pharmacies given both sets of findings.

I recommend that you accept this paper for publication following specialist statistical review.

Thank you very much for the positive feedback. We have now cited the Hann et al. paper, and we believe it enhances our discussion – thank you for highlighting the study to us.

Reviewer: 2

Reviewer Name: Jan Bauer

Institution and Country: Goethe University Frankfurt, Germany

Please state any competing interests or state 'None declared': None declared

Please leave your comments for the authors below

This manuscript examined clustering of pharmacies in England and showed that clustering exists and varies geographically and there is a positive association between clustering and social deprivation. The manuscript is well written and easy to read. However, there are some concerns regarding the methods and the context of this manuscript. Please see the comments below.

Thank you very much for the positive feedback. We will attempt to address your comments below.

Introduction:

The authors describe the potential of community pharmacies to fill the gap of GPs and AE departments in rural areas. However, there are obvious limitations for pharmacies filling that gap, which have not been addressed by the authors.

We agree and have amended our introduction to illustrate that community pharmacies can only fill the gap for certain things; more serious acute medical conditions should be managed by the GP or accident and emergency services, as appropriate.

The authors want to explore the relationship of clustering with urbanity. However is not explained why this has to be done. Please provide arguments as to why urbanity has to be included in the analysis.

Thank you for raising this point. We have now updated our introduction to state why urbanity was an important consideration in our analysis.

Essentially, our hypothesis was that, in rural areas, where the density of population is lower, there would be less community pharmacy clustering, when compared to urban areas. In view of this we felt

to was important to determine the clustering of community pharmacies according to urbanity, and we feel this is a strength of our paper. If we just included clustering 'overall' without considering urbanity it would not reflect a true account of the problem, and may bias our findings.

The authors mention the "positive care law" of pharmacies (access is highest in areas with highest deprivation). Later the authors declare that no study exist that examines the relationship of clustering and social deprivation. Access to healthcare can be measured in many different ways, also possibly by analyzing clusters. Therefore it is unclear if "clustering" is a measure of access. If so, please indicate this and provide context of other accessibility measures. If not please describe the difference between "clustering" and access and why this has to be analysed giving the already mentioned positive care law.

While we accept that "clustering" could be a measure of access, in this study, we do not consider community pharmacy clustering as a measure of accessibility, as we are not considering any measure of population in our analysis (i.e. people living in close proximity of the pharmacy; or people that are utilising the pharmacy). We are only interested in how each community pharmacy is located with respect to one another.

In contrast, for the "positive care law" community pharmacy access was measured according to the percentage of the population that lived within a community pharmacy by 20 minutes walking distance. Again, we do acknowledge that just because someone lives within a specific walking distance to a community pharmacy does not necessarily mean they are able – or would chose to access it.

We have updated the introduction to make it clear that the "positive care law" of community pharmacies considers population in the measure of access.

Methods:

The authors want to "explore geographical access to community pharmacies". However, again it is not clear what to authors mean by "access" (see also comment in the introduction) and how they want to measure access.

We agree with this point and believe the methods section was not clear. We have now updated this section to make it clearer and explain that we are not exploring accessibility as we are not considering any measures of population.

Please provide the year of the community pharmacy data.

We obtained our data from the open access Geo-healthcare access database; this used data on community pharmacy data from 2016. We have now included this statement in our methods section.

The authors used a 0,5 mile straight line buffer, which was supposed to represent "walking distance". Using straight-line buffers to measure walking distance is not state of the art in accessibility measures.

State of the art would be to use walking speed on a street network. It is unclear why the authors have not used a street network. Therefore the straight line buffer used should not be referred to as "10 minute walking distance" but to "0,5 mile radius" which is more precise and less misleading. It is also unclear why the authors defined a pharmacy cluster as 2 or more pharmacies within the buffer. Please explain the choice of "2 or more". It is also unclear why, for the deprivation analyses "3 or more" have been used for the definition of clusters.

We agree that measuring walking distance by straight line is not “state of the art”. We currently have a research project underway that will explore how people use community pharmacies, and then analyse users distance from the community pharmacy by walking distance, driving distance or public transport; all of these measures will be undertaken by street network. We hope to report our findings from this work in around 12 months.

The reason we included the walking distance by 0.5 mile radius, and explored clustering by 2 or more/3 or more was due to the policy implications of this work. Several policy documents consider walking distance by straight line, and clustering by 2 or more/3 or more, and we felt it was important to compare of our work to this. We have also acknowledged the limitations of this approach in our discussion under the limitations section.

Discussion:

Within the discussion the authors focused mainly on the possible implications. The actual discussion of the result in regard to other studies is almost completely missing. Furthermore putting the results in an international perspective would further improve this manuscript.

References:

Almost 50% of the presented literature (regarding journal articles) are form the authors themselves. Given the topic of this manuscript and the vast (international) available literature, the choice of the literature seems biased.

The main emphasis (and strength) of this paper was for the policy implications. We do accept that we have could have further discussed our findings with regards to other studies. Although we are unable to find other studies that have specifically explored the clustering of community pharmacies, we have considered wider literature that explores accessibility and uptake of healthcare services by distance and urbanity; to increase the international interest of this work, we have cited studies from Canada and Ethiopia in our discussion.

While we accept that we have cited many of our previous research in this paper, this is largely due our track record and expertise in this area. We have been working on this programme of research for many years, and citing our previous work is important to put this work into context, and frame our paper. We would disagree that we have been biased in our selections, as given the novelty of this work, we have only cited the literature that is available. However, given that we have expanded our discussion to compare with wider literature, we have also expanded on the literature we have included in our manuscript.

VERSION 2 – REVIEW

REVIEWER	Jan Bauer Goethe University Frankfurt, Germany
REVIEW RETURNED	06-May-2018

GENERAL COMMENTS	All comments have been addressed accordingly. I suggest to accept the paper in its current form.
--

REVIEWER	Sally Jacobs The University of Manchester, UK
REVIEW RETURNED	07-May-2018

GENERAL COMMENTS	I have reviewed the changes to this paper and authors' responses in light of the reviewers' comments and I am satisfied that the concerns raised have been adequately addressed.
--